# Locally Weighted Discriminant Analysis for Hyperspectral Image Classification

**Xiaoyan Li [1,*], Lefei Zhang [2] and Jane You [3]**

[1] School of Computer Science, China University of Geosciences, Wuhan 430074, China
[2] School of Computer, Wuhan University, Wuhan 430072, China; zhanglefei@whu.edu.cn
[3] Department of Computing, The Hong Kong Polytechnic University, Kowloon, Hong Kong 999077, China; csyjia@comp.polyu.edu.hk
[*] Correspondence: lixy@cug.edu.cn; Tel.: +86-27-6788-3716

**Abstract:** A hyperspectral image (HSI) contains a great number of spectral bands for each pixel, which will limit the conventional image classification methods to distinguish land-cover types of each pixel. Dimensionality reduction is an effective way to improve the performance of classification. Linear discriminant analysis (LDA) is a popular dimensionality reduction method for HSI classification, which assumes all the samples obey the same distribution. However, different samples may have different contributions in the computation of scatter matrices. To address the problem of feature redundancy, a new supervised HSI classification method based on *locally weighted discriminant analysis* (LWDA) is presented. The proposed LWDA method constructs a weighted discriminant scatter matrix model and an optimal projection matrix model for each training sample, which is on the basis of discriminant information and spatial-spectral information. For each test sample, LWDA searches its nearest training sample with spatial information and then uses the corresponding projection matrix to project the test sample and all the training samples into a low-dimensional feature space. LWDA can effectively preserve the spatial-spectral local structures of the original HSI data and improve the discriminating power of the projected data for the final classification. Experimental results on two real-world HSI datasets show the effectiveness of the proposed LWDA method compared with some state-of-the-art algorithms. Especially when the data partition factor is small, i.e., 0.05, the overall accuracy obtained by LWDA increases by about 20% for *Indian Pines* and 17% for *Kennedy Space Center (KSC)* in comparison with the results obtained when directly using the original high-dimensional data.

**Keywords:** hyperspectral image (HSI) classification; linear discriminant analysis (LDA); dimensionality reduction; spatial-spectral information

## 1. Introduction

A hyperspectral image (HSI) is captured by an image spectrometer with hundreds of spectral bands for each image pixel, which often plays an important role in the fields of urban planning, precision agriculture, and land-cover classification [1–5]. Generally, the spectral bands of each pixel are considered to be the features with high dimensionality. The high dimensionality of the original HSI data significantly leads to feature redundancy problem and increases the computational complexity [6,7]. To overcome these drawbacks, it is critical to perform dimensionality reduction, which is designed to project the original high-dimensional data into a low-dimensional feature subspace while preserving some desirable information.

The existing dimensionality reduction approaches can be classified into two categories: feature selection [8–11] and feature extraction. The focus of this paper is feature extraction, which is designed

to construct a low-dimensional embedding subspace and then create meaningful information by the projection of the original high-dimensional data. Then, the existing traditional classification methods (e.g., support vector machine classifier) can be directly applied to the projected data. Therefore, the HSI classification of low-dimensional data is conducive to avoiding feature redundancy and the Hughes phenomenon [12] and to reducing the computational complexity. Consequently, lots of feature extraction approaches have been presented [6,13–20]. Popular feature extraction methods include principal component analysis (PCA) [21], linear discriminant analysis (LDA) [22], locality preserving projection (LPP) [23], and modified locality preserving projection (MLPP) [24]. Compared with PCA and LPP, LDA can learn a linear transformation by simultaneously minimizing the intraclass distances and maximizing the interclass discrepancy. However, when directly applying LDA to process HSI data, it still faces several problems [25]: (1) when the dimensionality of data exceeds the size of training samples, LDA suffers from an ill-posed problem; (2) when the reduced dimensionality is less than the number of classes, LDA has an over-reducing problem; (3) LDA neglects the spatial information in the discriminant analysis; (4) LDA assumes that all the samples obey the Gaussian distribution, which is difficult for constructing the local classification boundary.

Recently, there have been many variants of LDA that try to improve the classification performance using some constraints, such as regularized local discriminant embedding (RLDE) [26], local geometric structure Fisher analysis (LGSFA) [18], and locality adaptive discriminant analysis (LADA) [27]. RLDE employs a regularized discriminant model to preserve the local structure of the HSI data. LGSFA retains the local structure among the within-class and between-class samples during the analysis process. LADA constructs a scatter matrix for each pixel with its small neighborhood, which is considered a regularization term. The above approaches can alleviate the ill-posed and over-reducing problems of the original LDA method. However, they only represent the local structure relationship of the HSI data as one-to-one. Moreover, the preservation of local structure still remains an open issue. Some new methods have been developed on the basis of graph learning. Ly et al. [28] used graph learning to construct scatter matrices, and then they conducted a discriminant analysis. Li et al. [29] proposed a two-stage framework to learn the data graph in the low-dimensional feature subspace. However, the above methods only consider the spectral information of the HSI data, which cannot accurately determine the local classification boundary.

To effectively exploit the spectral information and spatial information of the high-dimensional HSI data, a *locally weighted discriminant analysis* (LWDA)-based dimensionality reduction method is proposed for HSI classification in this paper. In order to apply the spatial information to the projection process, the proposed method learns the data structures adaptively during the transformation of subspace projection. Furthermore, to guarantee the spatial consistency of land cover, samples within a small neighborhood in the embedding space should be similar, which is considered a regularized constraint term during the optimization. The main contributions of this paper can be summarized as follows:

(1)　A weighted scatter matrix model is proposed by exploiting the label information and spectral information of the samples, which is able to reduce the effect of the image difference of the HSI data.
(2)　The proposed method considers the spatial consistency and the similarity relationship among the samples in a small spatial neighborhood, which is able to describe the local structure of the samples.
(3)　An optimization function is constructed on the basis of the spatial-spectral information and label information, which is able to preserve the within-class characteristics and suppress the between-class properties in the embedding feature subspace.

The remainder of this paper is organized as follows. Section 2 briefly introduces some related works, including the original LDA and MFA approaches. Section 3 provides our proposed method in detail. In Section 4.2, experimental results are presented to demonstrate the effectiveness of the proposed method compared with several state-of-the-art dimensionality reduction algorithms. Finally, a conclusion of this work is provided in Section 5.

## 2. Related Works

Let $X = [x_1, x_2, \cdots, x_n] \in \mathbb{R}^{d \times n}$ be the original HSI data, where $d$ is the number of spectral bands for each image pixel, i.e., the data dimensionality of the HSI data, and $n$ represents the number of the image pixels considered as samples. The label information of the $i$th pixel is denoted as $\ell(x_i)$, which belongs to $\{1, 2, \cdots, c\}$, and $c$ is the number of classes. The goal of dimensionality reduction is to construct a projection matrix $P \in \mathbb{R}^{d \times m}$, where $m$ is the reduced dimensionality of the projected data. For the linear mapping function, the projected data is indicted as $Y = P^T X$. Generally, the value of $m$ is considerably smaller than $d$.

### 2.1. Linear Discriminant Analysis

Linear discriminant analysis (LDA) is a supervised method and able to compact the within-class samples and separate the between-class samples. It defines a between-class scatter matrix $S_b$ and a within-class scatter matrix $S_w$ as follows:

$$S_b = \sum_{k=1}^{c} n_k \left( u_k - \bar{u} \right) \left( u_k - \bar{u} \right)^T, \tag{1}$$

$$S_w = \sum_{k=1}^{c} \sum_{i=1}^{n_k} \left( x_k^i - u_k \right) \left( x_k^i - u_k \right)^T, \tag{2}$$

where $n_k$ is the number of the $k$th class, and $x_k^i$ is the $i$th sample from the $k$th class. $u_k$ is the mean of the $k$th class, computed by $u_k = \sum_{i=1}^{n_k} x_k^i / n_k$. Similarly, $\bar{u}$ is the mean of all the samples, i.e., $\bar{u} = \sum_{k=1}^{c} \sum_{i=1}^{n_k} x_k^i / \sum_{k=1}^{c} n_k$. In addition, $T$ represents the transpose operation.

With the above definitions, LDA tries to learn the linear transformation matrix $P$ by maximizing the ratio of the between-class scatter and the within-class scatter. The projection matrix can be obtained by the following optimization function [22]:

$$\max_{P} tr \left( \frac{P^T S_b P}{P^T S_w P} \right), \tag{3}$$

where $tr(\cdot)$ represents the trace operator. The optimal projection matrix $P^\star$ can be obtained by analytically solving the generalized eigenvalue decomposition and then choosing the $m$ eigenvectors that correspond to the $m$ largest eigenvalues. Then, the $m$-dimensional projected data can be computed by $Y = (P^\star)^T X$.

Equations (1) and (2) reveal that the between-class scatter matrix is easily reflected by the subtraction of the total mean. Moreover, it is unable to capture the local manifold structure of the HSI data. Due to the two drawbacks, it is difficult for LDA to achieve satisfactory performance in real-world HSI applications.

### 2.2. Marginal Fisher Analysis

Marginal Fisher analysis (MFA) is a supervised graph learning method, which constructs an inherent graph and a penalty graph [18]. The inherent graph tries to obtain certain geometrical information of the input dataset, while the penalty graph reveals the unwanted properties of the inputs. MFA designs two weight matrices. Let $W = \{w_{ij}\}_{i,j=1}^{n}$ and $W^p = \{w_{ij}^p\}_{i,j=1}^{n}$ be the similarity matrix and the penalty matrix. $w_{ij}$ represents the similarity relationship between the two data points $x_i$ and $x_j$, which are from the same class. On the other hand, $w_{ij}^p$ describes the similarity characteristic between $x_i$ and $x_j$ that are from different classes. The mathematical descriptions of $w_{ij}$ and $w_{ij}^p$ are defined as follows:

$$w_{i,j} = \begin{cases} 1, & if \ (\boldsymbol{x}_i \in \mathcal{N}_1(\boldsymbol{x}_j) \ or \ \boldsymbol{x}_j \in \mathcal{N}_1(\boldsymbol{x}_i)) \\ & and \ \ell(\boldsymbol{x}_i) = \ell(\boldsymbol{x}_j), \\ 0, & otherwise. \end{cases} \tag{4}$$

$$w_{i,j}^p = \begin{cases} 1, & if \ (\boldsymbol{x}_i \in \mathcal{N}_2(\boldsymbol{x}_j) \ or \ \boldsymbol{x}_j \in \mathcal{N}_2(\boldsymbol{x}_i)) \\ & and \ \ell(\boldsymbol{x}_i) \neq \ell(\boldsymbol{x}_j), \\ 0, & otherwise. \end{cases} \tag{5}$$

where $\mathcal{N}_1(\boldsymbol{x}_i)$ and $\mathcal{N}_1(\boldsymbol{x}_j)$ represent the $k_1$ nearest neighbors of data points $\boldsymbol{x}_i$ and $\boldsymbol{x}_j$ that are from the same class, and $\mathcal{N}_2(\boldsymbol{x}_i)$ and $\mathcal{N}_2(\boldsymbol{x}_j)$ represent the $k_2$ nearest neighbors of data points $\boldsymbol{x}_i$ and $\boldsymbol{x}_j$ that are from different classes, respectively.

With the definition of the two weight matrices, the optimization function of MFA is designed to obtain a projection matrix $\boldsymbol{P}$ as follows:

$$\min_{\boldsymbol{P}} \frac{\sum_{i=1}^n \sum_{j=1}^n \left\| \boldsymbol{P}^T(\boldsymbol{x}_i - \boldsymbol{x}_j) \right\|_2^2 w_{ij}}{\sum_{i=1}^n \sum_{j=1}^n \left\| \boldsymbol{P}^T(\boldsymbol{x}_i - \boldsymbol{x}_j) \right\|_2^2 w_{ij}^p}, \tag{6}$$

$$\Rightarrow \min_{\boldsymbol{P}} tr \left( \frac{\boldsymbol{P}^T \boldsymbol{X} \boldsymbol{L} \boldsymbol{X}^T \boldsymbol{P}}{\boldsymbol{P}^T \boldsymbol{X} \boldsymbol{L}^p \boldsymbol{X}^T \boldsymbol{P}} \right), \tag{7}$$

where $\boldsymbol{L}$ and $\boldsymbol{L}^p$ are the Laplacian matrices, which are defined as $\boldsymbol{L} = \boldsymbol{D} - \boldsymbol{W}, \boldsymbol{D} = diag(\{\sum_{j=1}^n w_{ij}\}_{i=1}^n)$, and $\boldsymbol{L}^p = \boldsymbol{D}^p - \boldsymbol{W}^p$, $\boldsymbol{D}^p = diag(\{\sum_{j=1}^n w_{ij}^p\}_{i=1}^n)$. $diag(\cdot)$ represents the matrix diagonal element extraction operation.

Equation (7) can be solved analytically through the generalized eigenvalue decomposition of $\boldsymbol{X} \boldsymbol{L} \boldsymbol{X}^T$ and $\boldsymbol{X} \boldsymbol{L}^p \boldsymbol{X}^T$. Then, the optimal projection matrix $\boldsymbol{P}^\star$ is formed by the $m$ eigenvectors corresponding to the $m$ smallest eigenvalues.

MFA tries to enhance the compactness of the data points from the same class and to improve the separability of the data points from different classes in the embedding feature subspace. However, the similarity relationship between two data points in a small neighborhood is simplified as 1, which will limit it to learn a certain local manifold structure of the HSI data.

## 3. Proposed Method

To take advantage of the discriminant information and the spatial-spectral information of the input HSI data, a new supervised dimensionality reduction method, named *locally weighted discriminant analysis* (LWDA), is presented for HSI classification. LWDA constructs a weighted scatter matrix model on the basis of the within-class and between-class scatter matrices of the traditional LDA method. The weighted scatter matrix model defines a weighted within-class scatter matrix and a weighted between-class scatter matrix to improve the discriminating power. Furthermore, LWDA preserves the spatial consistency among the samples in a small spatial neighborhood. To construct the optimal low-dimensional feature subspace, the proposed method obtains the corresponding projection matrix by compacting the nature of the weighted within-class scatter matrix and the spatial consistency, and suppressing the property of the weighted between-class scatter matrix.

The flowchart of the proposed method is shown in Figure 1, where the high-dimensional HSI data are projected onto a two-dimensional subspace for visualization. Taking the classification process of the *Indian Pines* dataset as an example, the steps of the proposed algorithm can be summarized as follows: (1) on the basis of the training samples and the corresponding training labels, the weighted within- and between-class scatter matrices are computed; (2) with the help of the training and test samples, the spatial consistency matrix for each training sample can be computed; (3) with the foregoing weighted scatter matrices and spatial consistency matrix, the optimal projection matrix corresponding to each training sample is obtained; (4) for each test sample, the spatially closest training sample's

projection matrix can be obtained, which is used to construct the embedding features; (5) the class estimation of each test sample is obtained by a certain classifier with the training labels and the embedding features.

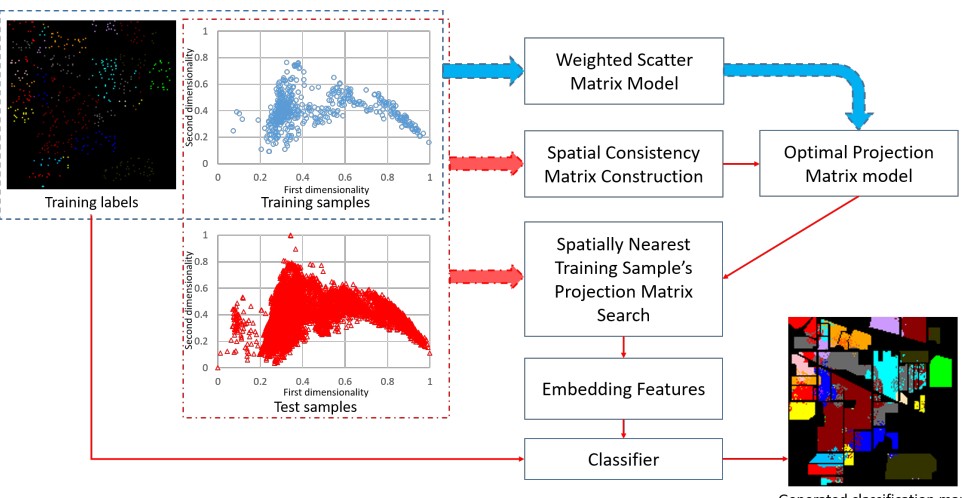

**Figure 1.** Flowchart of the proposed LWDA method. With a partition factor $\tau$, the training and test sample set (shown in two-dimensional space), as well as the training label set, are obtained. Then, the dataset is used to construct the weighted scatter matrix model, spatial consistency matrix, and optimal projection matrix for each training sample. The next step is to find the spatially nearest training sample's projection matrix, which is applied to construct the embedding features of training samples and input test sample. Once all the predicted test labels are obtained, the classification map (including the training labels), is generated by exploiting a fixed classifier.

### 3.1. Weighted Scatter Matrix Model

LDA assumes all the samples possess the same contribution, i.e., the Gaussian distribution. In LDA, the within-class scatter matrix only considers the data variances of the within-class samples, while the between-class scatter matrix just considers the data variance between the mean of each individual class and the total mean. However, different within-class samples should have different contribution rates in the within-class scatter matrix. Moreover, the properties of any two different individual class means may be different in the between-class scatter matrix. To better represent the similarity characteristic of the within-class samples and different individual class means, the proposed method constructs two weighted scatter matrices, i.e., the weighted within-class scatter matrix and the weighted between-class scatter matrix, which are defined as follows:

$$\widetilde{\boldsymbol{S}}_w = \sum_{k=1}^{c} \sum_{i=1}^{n_k} \sum_{j=1}^{n_k} \left(\boldsymbol{x}_k^i - \boldsymbol{u}_k\right) g_{i,j}^k \left(\boldsymbol{x}_k^j - \boldsymbol{u}_k\right)^T , \tag{8}$$

$$\widetilde{\boldsymbol{S}}_b = \sum_{i=1}^{c} \sum_{j=1}^{c} n_i \left(\boldsymbol{u}_i - \boldsymbol{u}_j\right) h_{i,j} \left(\boldsymbol{u}_i - \boldsymbol{u}_j\right)^T , \tag{9}$$

where $g_{i,j}^k$ is the similarity weight between the samples $\boldsymbol{x}_k^i$ and $\boldsymbol{x}_k^j$, and $h_{i,j}$ is the similarity weight between the one-class means $\boldsymbol{u}_i$ and $\boldsymbol{u}_j$. The similarity weights are represented as

$$g_{i,j}^k = exp\left(-\frac{\left\|\boldsymbol{x}_k^i - \boldsymbol{x}_k^j\right\|^2}{2\left(\rho_k^i\right)^2 + \varepsilon}\right) , \tag{10}$$

$$h_{i,j} = exp\left(-\frac{\|\boldsymbol{u}_i - \boldsymbol{u}_j\|^2}{2\sigma_i^2 + \varepsilon}\right), \tag{11}$$

where $\rho_k^i = \sum_{j=1}^{n_k} \left\|\boldsymbol{x}_k^i - \boldsymbol{x}_k^j\right\| / n_k$, $\sigma_i = \sum_{j=1}^{c} \|\boldsymbol{u}_i - \boldsymbol{u}_j\| / c$, and $\varepsilon$ is a small value for avoiding zero in the denominator.

Similar to LDA, the optimization function is designed to improve the aggregation of the within-class samples and enhance the diversity of the between-class samples in a low-dimensional feature subspace. So, the optimal projection matrix can be obtained by the following formula:

$$\min_{\boldsymbol{P}} tr\left(\frac{\boldsymbol{P}^T \widetilde{\boldsymbol{S}}_w \boldsymbol{P}}{\boldsymbol{P}^T \widetilde{\boldsymbol{S}}_b \boldsymbol{P}}\right). \tag{12}$$

Supposing the minimum value of the above function is $\alpha$, the optimal $\boldsymbol{P}$ should make the value of $tr\left(\boldsymbol{P}^T \widetilde{\boldsymbol{S}}_w \boldsymbol{P}\right) - \alpha tr\left(\boldsymbol{P}^T \widetilde{\boldsymbol{S}}_b \boldsymbol{P}\right)$ close to 0. Thus, Equation (12) is equivalent to

$$\min_{\boldsymbol{P}} tr\left(\boldsymbol{P}^T \widetilde{\boldsymbol{S}}_w \boldsymbol{P}\right) - \alpha tr\left(\boldsymbol{P}^T \widetilde{\boldsymbol{S}}_b \boldsymbol{P}\right). \tag{13}$$

### 3.2. Spatial Consistency Matrix

For real-world HSI data, the data points within a small spatial region are often highly correlated and are classified as the same class [25]. Hence, spatial consistency is essential for an accurate classification. Given a data point $\boldsymbol{x}_i \in \mathbb{R}^{d \times 1} (i = 1, \cdots, n)$, the spatial surroundings are found within a search region with a size of $r \times r$, where $r$ must be odd. Therefore, the $r^2 - 1$ neighbors are selected for each sample, which are denoted as $\boldsymbol{Z}_i = \left[\boldsymbol{z}_i^1, \boldsymbol{z}_i^2, \cdots, \boldsymbol{z}_i^{r^2-1}\right]$ and $\boldsymbol{z}_i^j \in \boldsymbol{X}(j = 1, \cdots, r^2 - 1)$. For different samples $\boldsymbol{x}_i$ and $\boldsymbol{x}_j$, the subsets $\boldsymbol{Z}_i$ and $\boldsymbol{Z}_j$ may partially overlap. In a desired feature subspace, these neighbors are encouraged to be close to each other. The problem of spatial consistency can be defined as

$$\min_{\boldsymbol{P}} \sum_{j,k=1}^{r^2-1} \left\|\boldsymbol{P}^T\left(\boldsymbol{z}_i^j - \boldsymbol{z}_i^k\right)\right\|_2^2, \qquad \forall \boldsymbol{x}_i,\ i = 1, 2, \cdots, n. \tag{14}$$

The spatial consistency matrix is defined as

$$\widetilde{\boldsymbol{S}}_z = \sum_{j,k=1}^{r^2-1} \left(\boldsymbol{z}_i^j - \boldsymbol{z}_i^k\right)\left(\boldsymbol{z}_i^j - \boldsymbol{z}_i^k\right)^T. \tag{15}$$

Then, Equation (14) can be further reduced to

$$\min_{\boldsymbol{P}} tr\left(\boldsymbol{P}^T \widetilde{\boldsymbol{S}}_z \boldsymbol{P}\right). \tag{16}$$

### 3.3. Optimization Algorithm

Integrating Equations (13) and (16) together, the objective function of the proposed LWDA method is summarized as

$$\min_{\boldsymbol{P}_i} tr\left(\boldsymbol{P}_i^T \widetilde{\boldsymbol{S}}_w \boldsymbol{P}_i\right) - \alpha tr\left(\boldsymbol{P}_i^T \widetilde{\boldsymbol{S}}_b \boldsymbol{P}_i\right) + \beta tr\left(\boldsymbol{P}_i^T \widetilde{\boldsymbol{S}}_z \boldsymbol{P}_i\right), \tag{17}$$

$$\Rightarrow \min_{\boldsymbol{P}_i} tr\left(\boldsymbol{P}_i^T \left(\widetilde{\boldsymbol{S}}_w - \alpha \widetilde{\boldsymbol{S}}_b + \beta \widetilde{\boldsymbol{S}}_z\right) \boldsymbol{P}_i\right), \tag{18}$$

where $\alpha$ and $\beta$ are parameters, and $\boldsymbol{P}_i$ is the desired projection matrix for the sample $\boldsymbol{x}_i (i = 1, 2, \cdots, n)$. With the proposed objective function, the spatial consistency between the data points is captured,

and the local data relationship is also investigated during the discriminant analysis. The optimal $P_i$ for Equation (18) can be obtained by the $m$ ($m \ll d$) eigenvectors of the critical matrix $\left( \widetilde{S}_w - \alpha \widetilde{S}_b + \beta \widetilde{S}_z \right)$ corresponding to the $m$ smallest eigenvalues.

With a certain dataset partition factor $\tau$ ($0 < \tau < 1$), the input HSI dataset can be divided into the training subset $X_s$ and the test subset $X_t$. That is to say, for the $k$th class, the number of the randomly selected samples for the training subset is $n_{s,k} = \lceil n_k * \tau \rceil$, while the number of the samples chosen for the test subset is $n_{t,k} = n_k - n_{s,k}$, where $n_k$ is the total number of samples belonging to the $k$th class. For simplicity, the training and test subsets are denoted as $X_s = \left[ x_s^1, x_s^2, \cdots, x_s^{n_s} \right]$ and $X_t = \left[ x_t^1, x_t^2, \cdots, x_t^{n_t} \right]$. The label information of the training subset is marked as $Y_s = \left[ y_s^1, y_s^2, \cdots, y_s^{n_s} \right]$, where $y_s^i \in [1, 2, \cdots, c]$, $i = 1, 2, \cdots, n_s$. The details of the whole framework are described in Algorithm 1.

---

**Algorithm 1:** Locally weighted discriminant analysis (LWDA).

---

**Input:** Training dataset $X_s$, training class information set $Y_s$, test dataset $X_t$, parameters $\alpha$ and $\beta$, dimensionality of desired projection matrix $m$.

**Output:** Estimate the test class information set $Y_t$.

**Training:**

1. Compute the weighted within-class scatter matrix $\widetilde{S}_w$ according to Equation (8);
2. Compute the weighted between-class scatter matrix $\widetilde{S}_b$ according to Equation (9);

**for** $i = 1, 2, \cdots, n_s$ **do**

3. Construct the neighbor set $Z_i$ and then compute the spatial consistency matrix $\widetilde{S}_z$ according to Equation (15);
4. Compute the total matrix $\widetilde{S} = \widetilde{S}_w - \alpha \widetilde{S}_b + \beta \widetilde{S}_z$;
5. Obtain the optimal projection matrix $P_i$ by choosing the $m$ eigenvectors of $\widetilde{S}$ corresponding to the $m$ smallest eigenvalues; then, set $i = i + 1$;

**end**

**Testing:**

**for** $i = 1, 2, \cdots, n_t$ **do**

6. For each test sample $x_t^i$, find the spatially nearest training sample $x_s^{i^\star}$ and the corresponding optimal projection matrix, denoted as $\widetilde{P}_i = P_{i^\star}$, where $i^\star = 1, \cdots, n_s$;
7. Low-dimensional embedding features are computed as $X_{s,m} = \widetilde{P}_i^T X_s$, and $x_{t,m}^i = \widetilde{P}_i^T x_t^i$.
8. Using the nearest-neighbor classifier, find the serial number of the nearest training sample in the low-dimensional feature subspace, computed as $j^\star = \arg \min_j \left\| x_{t,m}^i - x_{s,m}^j \right\|_2^2$, where $j = 1, \cdots, n_s$;
9. Obtain the corresponding class information, i.e., $y_t^i = y_s^{j^\star}$; then, set $i = i + 1$.

**end**

---

## 4. Experimental Results

### 4.1. Experimental Setting

In the experiments, two real-world hyperspectral image datasets were employed, i.e., *Indian Pines* and *Kennedy Space Center* (*KSC*) datasets [30]. The *Indian Pines* dataset contains 10,249 data points from 16 classes. Each data point (pixel) has 200 spectral bands. The *KSC* dataset annotates 5211 valid pixels (excluding the background pixels with the class information of 0) from 13 classes. Each pixel has

176 spectral bands. The two HSI datasets were both captured by an Airborne Visible/Infrared Imaging Spectrometer (AVIRIS) sensor. Table 1 shows various land-cover types and the corresponding number of samples for the two aforementioned HSI datasets.

**Table 1.** Number of total, training, and test samples, with a partition factor $\tau = 0.05$ of each land-cover class for the *Indian Pines* and *Kennedy Space Center (KSC)* datasets.

| | *Indian Pines* | | | | | *KSC* | | | |
|---|---|---|---|---|---|---|---|---|---|
| Class No. | Land Cover | Samples | Training | Test | Class No. | Land Cover | Samples | Training | Test |
| 1 | Alfalfa | 46 | 3 | 43 | 1 | Scurb | 761 | 39 | 722 |
| 2 | Corn-notill | 1428 | 72 | 1356 | 2 | Willow-swamp | 243 | 13 | 230 |
| 3 | Corn-mintill | 830 | 42 | 788 | 3 | Cabbage-palm-hammock | 256 | 13 | 243 |
| 4 | Corn | 237 | 12 | 225 | 4 | Cabbage-palm/oak-hammock | 252 | 13 | 239 |
| 5 | Grass-pasture | 483 | 25 | 458 | 5 | Slash-pine | 161 | 9 | 152 |
| 6 | Grass-tree | 730 | 37 | 693 | 6 | Oak/broadleaf-hammock | 229 | 12 | 217 |
| 7 | Grass-pasture-mowed | 28 | 2 | 26 | 7 | Hardwood-swamp | 105 | 6 | 99 |
| 8 | Hay-windrowed | 478 | 24 | 454 | 8 | Graminoid-marsh | 431 | 22 | 409 |
| 9 | Oats | 20 | 1 | 19 | 9 | Spartina-marsh | 520 | 26 | 494 |
| 10 | Soybeans-notill | 972 | 49 | 923 | 10 | Cattail-marsh | 404 | 21 | 383 |
| 11 | Soybeans-mintill | 2455 | 123 | 2332 | 11 | Salt-marsh | 419 | 21 | 398 |
| 12 | Soybeans-clean | 593 | 30 | 563 | 12 | Mud-flats | 503 | 26 | 477 |
| 13 | Wheat | 205 | 11 | 194 | 13 | Water | 927 | 47 | 880 |
| 14 | Woods | 1265 | 64 | 1201 | | | | | |
| 15 | Bldg-grass-tree-drives | 386 | 20 | 366 | | | | | |
| 16 | Stone-steel-towers | 93 | 5 | 88 | | | | | |

To investigate the classification performance, each dataset was randomly divided into the training and test samples with a data partition factor $\tau$. For instance, the total number of samples for the "Corn-notill" land-cover type is 1428, shown in Table 1. Setting $\tau$ to 0.05, the number of samples chosen for training is $\lceil 1428 \times 0.05 \rceil = 72$, while the remaining 1356 samples were used for testing.

For a quantitative comparison, seven dimensionality reduction algorithms were taken as competitors, including the original raw spectral feature (RAW), principal component analysis (PCA) [21], linear discriminant analysis (LDA) [22], discrimination-information-based locality preserving projection (DLPP) [24], marginal Fisher analysis (MFA) and local geometric structure Fisher analysis (LGSFA) [18], two-stage subspace projection (TwoSP) [29], as well as discriminant analysis with graph learning (DAGL) [25]. In LWDA, the value of parameter $\alpha$ is set to $10^{-3}$.

According to different dimensionality reduction algorithms, the projection matrix can be obtained to achieve the low-dimensional embedding features of training and test samples. After that, a certain classifier, e.g., nearest neighbor (NN) and support vector machine (SVM) [31], is exploited to discriminate the land-cover types of the test samples with the help of the class information of training samples. In this study, three widely used classification measurements, i.e., average classification accuracy of all the classes (AA), overall classification accuracy (OA), and kappa coefficient (KC), were used to evaluate the objective results of each method. To alleviate the random error caused by the partition of training and test samples, each experiment was repeated five times in each condition; reported are the average AAs, the average OAs, the average KCs, and their standard deviations (STDs). All the experiments were performed on a personal computer with Intel Xeon CPU E5-2643 v3, 3.40 GHz, 64 GB memory, and 64-bit Windows 7 using Matlab R2017b.

*4.2. Performance on Hyperspectral Image Datasets*

The quantitative results of the proposed method and the baselines are given in Tables 2 and 3 for the *Indian Pines* dataset and *KSC* dataset, respectively. The two tables show the average classification accuracy of each class, the AAs, OAs, and KCs, as well as their STDs, which were obtained by repeating each experiment five times. All the values are represented in terms of percentage.

**Table 2.** Classification results (%) of each class ($\tau = 0.05$) with nearest-neighbor (NN) classifier on the *Indian Pines* dataset.

| Class No. | RAW | PCA | LDA | DLPP | MFA | LGSFA | TwoSP | DAGL | LWDA |
|---|---|---|---|---|---|---|---|---|---|
| 1 | $40.0 \pm 12.6$ | $42.3 \pm 20.6$ | $42.8 \pm 17.8$ | $49.3 \pm 15.7$ | $50.2 \pm 11.7$ | $32.1 \pm 12.8$ | $31.6 \pm 12.2$ | $88.4 \pm 10.5$ | $90.7 \pm 8.4$ |
| 2 | $48.1 \pm 1.2$ | $49.3 \pm 2.0$ | $57.0 \pm 2.1$ | $59.8 \pm 3.1$ | $59.1 \pm 2.6$ | $61.3 \pm 2.4$ | $67.0 \pm 1.7$ | $75.2 \pm 2.6$ | $81.2 \pm 2.8$ |
| 3 | $44.2 \pm 2.4$ | $44.0 \pm 2.3$ | $42.6 \pm 6.6$ | $48.6 \pm 5.4$ | $44.8 \pm 3.5$ | $49.4 \pm 3.7$ | $56.1 \pm 3.2$ | $73.2 \pm 5.9$ | $77.2 \pm 7.8$ |
| 4 | $30.2 \pm 6.3$ | $28.2 \pm 6.9$ | $27.9 \pm 3.2$ | $31.8 \pm 5.3$ | $29.6 \pm 4.1$ | $29.3 \pm 4.9$ | $40.1 \pm 4.8$ | $75.1 \pm 6.2$ | $76.6 \pm 10.2$ |
| 5 | $76.3 \pm 5.1$ | $75.9 \pm 3.6$ | $83.6 \pm 5.5$ | $83.4 \pm 3.3$ | $83.6 \pm 3.2$ | $84.5 \pm 2.6$ | $84.1 \pm 4.4$ | $84.5 \pm 2.3$ | $86.7 \pm 1.2$ |
| 6 | $92.2 \pm 1.9$ | $92.4 \pm 2.2$ | $91.2 \pm 2.1$ | $90.1 \pm 2.5$ | $92.9 \pm 0.9$ | $91.2 \pm 2.3$ | $92.3 \pm 1.7$ | $84.6 \pm 2.2$ | $87.0 \pm 2.6$ |
| 7 | $80.0 \pm 10.0$ | $79.2 \pm 10.4$ | $76.2 \pm 14.7$ | $85.4 \pm 9.2$ | $85.4 \pm 5.0$ | $53.8 \pm 28.4$ | $82.3 \pm 6.4$ | $3.8 \pm 15.9$ | $24.4 \pm 17.8$ |
| 8 | $94.5 \pm 2.3$ | $94.9 \pm 2.5$ | $96.0 \pm 1.4$ | $93.7 \pm 2.8$ | $93.9 \pm 3.2$ | $96.7 \pm 1.7$ | $89.3 \pm 5.8$ | $98.5 \pm 1.3$ | $99.0 \pm 0.6$ |
| 9 | $14.7 \pm 10.8$ | $14.7 \pm 10.8$ | $12.6 \pm 9.6$ | $24.2 \pm 13.2$ | $20.0 \pm 11.4$ | $17.9 \pm 11.5$ | $30.5 \pm 9.4$ | $1.5 \pm 9.8$ | $5.3 \pm 9.1$ |
| 10 | $61.3 \pm 6.5$ | $62.2 \pm 6.7$ | $46.8 \pm 3.8$ | $57.4 \pm 2.9$ | $54.7 \pm 6.3$ | $52.9 \pm 3.1$ | $69.1 \pm 1.4$ | $65.0 \pm 8.1$ | $74.8 \pm 7.7$ |
| 11 | $67.4 \pm 3.4$ | $67.4 \pm 2.1$ | $64.8 \pm 1.7$ | $63.1 \pm 1.1$ | $71.1 \pm 4.9$ | $69.2 \pm 1.8$ | $74.8 \pm 2.0$ | $85.9 \pm 1.7$ | $88.4 \pm 1.4$ |
| 12 | $35.0 \pm 2.6$ | $34.6 \pm 3.7$ | $46.8 \pm 3.9$ | $48.4 \pm 4.9$ | $43.9 \pm 4.8$ | $49.8 \pm 4.6$ | $56.0 \pm 6.3$ | $76.2 \pm 5.5$ | $76.0 \pm 6.8$ |
| 13 | $93.3 \pm 1.5$ | $93.2 \pm 1.7$ | $92.6 \pm 3.7$ | $93.7 \pm 5.0$ | $95.2 \pm 3.5$ | $97.3 \pm 1.3$ | $96.4 \pm 1.3$ | $93.3 \pm 2.0$ | $81.3 \pm 11.9$ |
| 14 | $90.3 \pm 2.9$ | $89.3 \pm 3.8$ | $93.4 \pm 1.6$ | $94.0 \pm 2.0$ | $94.0 \pm 1.9$ | $95.0 \pm 1.2$ | $95.2 \pm 2.1$ | $96.2 \pm 1.3$ | $97.7 \pm 0.7$ |
| 15 | $27.2 \pm 2.0$ | $27.3 \pm 2.3$ | $44.6 \pm 8.1$ | $44.5 \pm 7.7$ | $43.3 \pm 6.9$ | $48.4 \pm 8.0$ | $46.7 \pm 6.1$ | $86.9 \pm 7.3$ | $92.7 \pm 1.1$ |
| 16 | $86.4 \pm 3.0$ | $86.4 \pm 3.1$ | $79.8 \pm 8.3$ | $82.1 \pm 6.0$ | $81.1 \pm 4.4$ | $84.3 \pm 4.9$ | $86.4 \pm 3.2$ | $45.5 \pm 25.1$ | $59.8 \pm 22.0$ |
| AA | $61.3 \pm 1.4$ | $61.3 \pm 2.0$ | $62.4 \pm 2.7$ | $65.6 \pm 0.8$ | $65.2 \pm 1.0$ | $63.3 \pm 2.9$ | $68.6 \pm 0.9$ | $70.9 \pm 2.9$ | $74.9 \pm 1.8$ |
| OA | $64.8 \pm 1.0$ | $64.9 \pm 0.8$ | $65.8 \pm 1.7$ | $67.5 \pm 0.9$ | $68.6 \pm 1.4$ | $69.3 \pm 1.1$ | $73.9 \pm 1.2$ | $81.7 \pm 1.2$ | $85.1 \pm 1.0$ |
| KC | $59.7 \pm 1.1$ | $59.8 \pm 0.8$ | $60.8 \pm 2.1$ | $62.9 \pm 1.0$ | $63.9 \pm 1.5$ | $64.7 \pm 1.3$ | $70.1 \pm 1.4$ | $79.2 \pm 1.5$ | $83.0 \pm 1.1$ |

**Table 3.** Classification results (%) of each class ($\tau = 0.05$) with NN classifier on the *KSC* dataset.

| Class No. | RAW | PCA | LDA | DLPP | MFA | LGSFA | TwoSP | DAGL | LWDA |
|---|---|---|---|---|---|---|---|---|---|
| 1 | $87.7 \pm 3.2$ | $87.6 \pm 3.2$ | $86.1 \pm 4.2$ | $82.4 \pm 1.8$ | $91.3 \pm 2.0$ | $85.7 \pm 2.3$ | $90.6 \pm 2.3$ | $94.3 \pm 1.1$ | $98.6 \pm 1.3$ |
| 2 | $75.7 \pm 13.3$ | $75.7 \pm 13.2$ | $86.4 \pm 1.9$ | $87.8 \pm 2.4$ | $88.7 \pm 2.8$ | $90.0 \pm 3.1$ | $77.4 \pm 2.0$ | $85.2 \pm 5.8$ | $84.0 \pm 12.7$ |
| 3 | $66.7 \pm 3.7$ | $66.8 \pm 3.8$ | $53.4 \pm 7.1$ | $55.3 \pm 7.7$ | $60.2 \pm 6.1$ | $58.4 \pm 4.7$ | $77.4 \pm 3.5$ | $78.2 \pm 4.2$ | $93.2 \pm 5.1$ |
| 4 | $50.0 \pm 3.3$ | $49.9 \pm 3.0$ | $39.9 \pm 4.5$ | $39.3 \pm 5.1$ | $45.0 \pm 9.5$ | $47.0 \pm 4.6$ | $64.0 \pm 4.2$ | $76.6 \pm 5.1$ | $85.1 \pm 7.0$ |
| 5 | $45.8 \pm 13.1$ | $45.4 \pm 12.9$ | $50.4 \pm 4.7$ | $51.3 \pm 1.9$ | $48.2 \pm 4.2$ | $46.2 \pm 5.0$ | $73.0 \pm 1.8$ | $88.8 \pm 3.2$ | $88.9 \pm 8.2$ |
| 6 | $34.2 \pm 5.2$ | $34.5 \pm 5.1$ | $50.6 \pm 8.2$ | $52.9 \pm 8.8$ | $34.6 \pm 7.1$ | $43.3 \pm 8.7$ | $59.0 \pm 7.5$ | $75.1 \pm 8.4$ | $100.0 \pm 0.0$ |
| 7 | $64.0 \pm 11.2$ | $63.0 \pm 10.3$ | $53.7 \pm 13.0$ | $47.3 \pm 9.2$ | $52.7 \pm 11.5$ | $44.2 \pm 15.2$ | $78.8 \pm 9.5$ | $98.0 \pm 2.9$ | $100.0 \pm 0.0$ |
| 8 | $69.4 \pm 6.8$ | $69.0 \pm 7.4$ | $78.4 \pm 4.4$ | $77.7 \pm 3.8$ | $81.6 \pm 6.2$ | $82.7 \pm 5.1$ | $83.1 \pm 3.5$ | $91.7 \pm 5.8$ | $93.3 \pm 6.4$ |
| 9 | $88.5 \pm 4.0$ | $88.5 \pm 4.0$ | $82.3 \pm 3.2$ | $80.7 \pm 2.3$ | $87.7 \pm 5.1$ | $84.3 \pm 4.4$ | $93.5 \pm 2.6$ | $97.8 \pm 3.2$ | $99.1 \pm 1.9$ |
| 10 | $81.1 \pm 2.4$ | $81.2 \pm 2.5$ | $94.8 \pm 0.8$ | $91.6 \pm 1.6$ | $94.2 \pm 2.5$ | $94.9 \pm 1.6$ | $82.5 \pm 1.7$ | $96.9 \pm 0.5$ | $100.0 \pm 0.0$ |
| 11 | $92.8 \pm 1.6$ | $92.8 \pm 1.6$ | $87.6 \pm 2.6$ | $87.5 \pm 3.1$ | $88.7 \pm 4.6$ | $84.9 \pm 5.5$ | $85.2 \pm 3.0$ | $96.2 \pm 1.3$ | $99.6 \pm 0.5$ |
| 12 | $78.3 \pm 4.7$ | $78.2 \pm 4.6$ | $89.1 \pm 2.6$ | $88.3 \pm 4.1$ | $82.2 \pm 3.9$ | $83.7 \pm 2.7$ | $80.9 \pm 4.2$ | $92.7 \pm 2.7$ | $98.0 \pm 1.5$ |
| 13 | $98.4 \pm 0.9$ | $98.4 \pm 0.9$ | $99.6 \pm 0.4$ | $98.5 \pm 0.9$ | $98.2 \pm 0.5$ | $98.8 \pm 0.7$ | $98.5 \pm 0.9$ | $98.1 \pm 0.2$ | $100.0 \pm 0.0$ |
| AA | $71.7 \pm 1.8$ | $71.6 \pm 1.8$ | $73.3 \pm 1.1$ | $72.4 \pm 1.3$ | $73.3 \pm 1.0$ | $72.6 \pm 1.6$ | $80.3 \pm 0.9$ | $90.0 \pm 0.7$ | $95.4 \pm 0.8$ |
| OA | $79.6 \pm 0.6$ | $79.5 \pm 0.6$ | $81.4 \pm 0.6$ | $80.3 \pm 0.6$ | $82.0 \pm 0.9$ | $81.2 \pm 1.5$ | $85.0 \pm 0.6$ | $92.3 \pm 0.8$ | $96.8 \pm 0.7$ |
| KC | $77.3 \pm 0.6$ | $77.2 \pm 0.6$ | $79.3 \pm 0.7$ | $78.1 \pm 0.7$ | $80.0 \pm 1.0$ | $79.0 \pm 1.7$ | $83.3 \pm 0.8$ | $91.4 \pm 0.7$ | $96.4 \pm 0.8$ |

Tables 2 and 3 demonstrate that the proposed LWDA method achieves better classification results in most classes compared with other methods, and it outperforms all the competitors in terms of AAs, OAs, and KCs. PCA neglects the nonlinear relationship from the original high-dimensional feature data, although it achieves dimensionality reduction. LDA and DLPP preserve the local manifold structure by exploiting the discrimination information of the training samples. However, they neglect the global structure in the dimensionality reduction. Since MFA simply considers the similarity relationship between two samples in a small neighborhood as one, it is difficult to learn an accurate local manifold structure. LGSFA can retain the local structure among the within- and between-class samples during the discriminant analysis. TwoSP preserves the global structure in the first-stage subspace projection, and it investigates the local structure of the HSI data adaptively. DAGL combines the spatial neighborhood information and data graph in the discriminant analysis process. However, the data graph is constructed in the original high-dimensional space, which still introduces data noise into the final projection process. The proposed LWDA method exploits the spectral and spatial information of the HSI data, and it then enhances the spatial consistency during the discriminant analysis. Therefore, LWDA produces the best classification performance on the two HSI datasets.

Moreover, the classification maps of the aforementioned methods on the *Indian Pines* and *KSC* datasets are shown in Figures 2 and 4. For better visualization, the local magnifications with a certain magnification factor are displayed in Figures 3 and 5. From Figures 2–5, the proposed LWDA method generates smoother classification maps and poses more homogeneous areas. LWDA enforces the

spectral-spatial information during the discriminant analysis. It not only preserves the global structure but also the local manifold structure with the help of the proposed weighted scatter matrix model and the construction of spatial consistency.

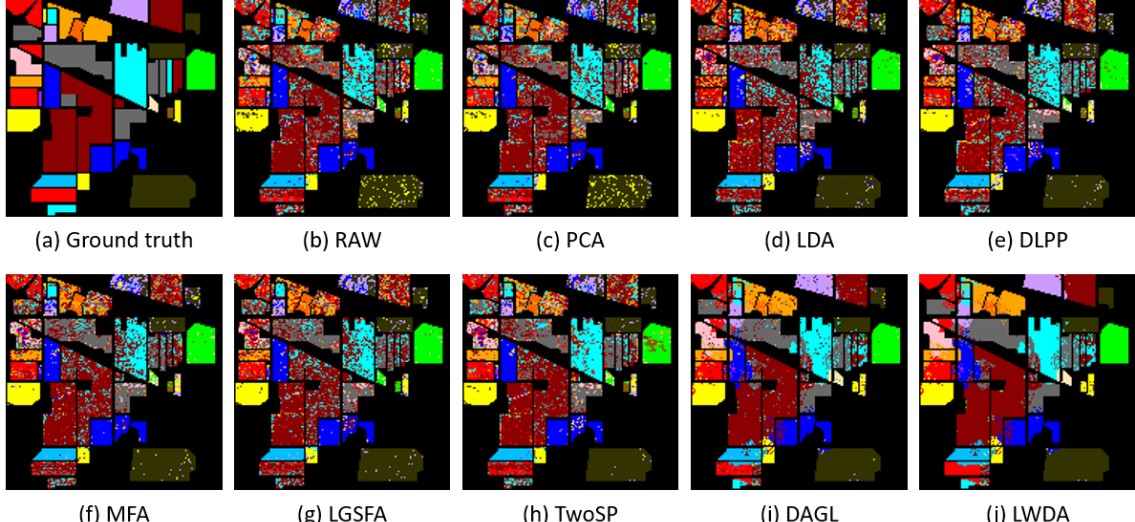

**Figure 2.** Classification maps of different dimensionality reduction methods with NN classifier on the *Indian Pines* dataset ($\tau = 0.05$). (**a**) Ground truth; (**b**) original raw spectral feature (RAW); (**c**) principal component analysis (PCA); (**d**) linear discriminant analysis (LDA); (**e**) discrimination-information-based locality preserving projection (DLPP); (**f**) marginal Fisher analysis (MFA); (**g**) local geometric structure Fisher analysis (LGSFA); (**h**) two-stage subspace projection (TwoSP); (**i**) discriminant analysis with graph learning (DAGL); (**j**) proposed locally weighted discriminant analysis (LWDA).

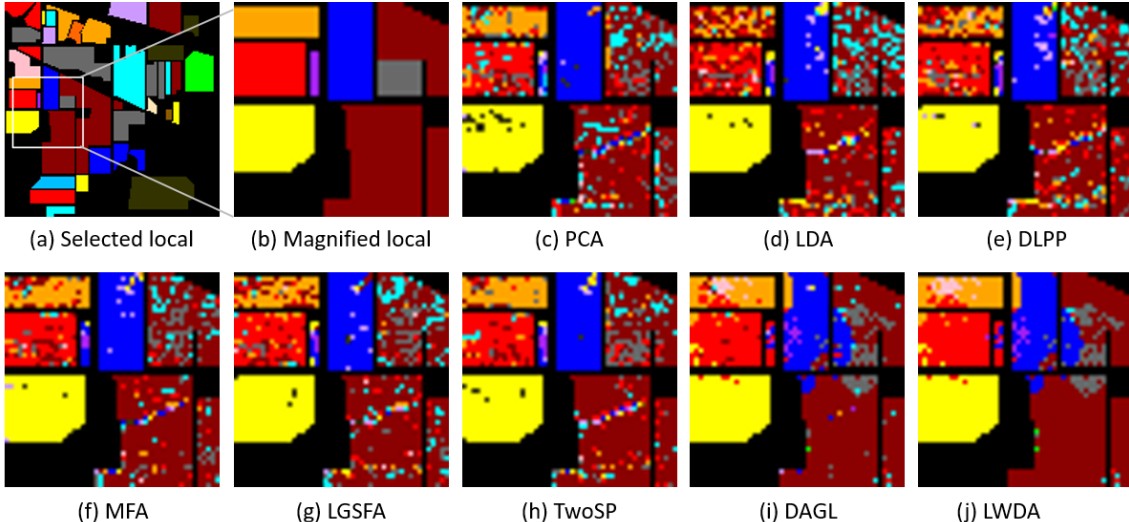

**Figure 3.** Local magnification (with a magnification factor of 3) of the output classification maps shown in Figure 2. (**a**) Selected local region; (**b**) magnified local region; (**c**) PCA; (**d**) LDA; (**e**) DLPP; (**f**) MFA; (**g**) LGSFA; (**h**) TwoSP; (**i**) DAGL; (**j**) proposed LWDA.

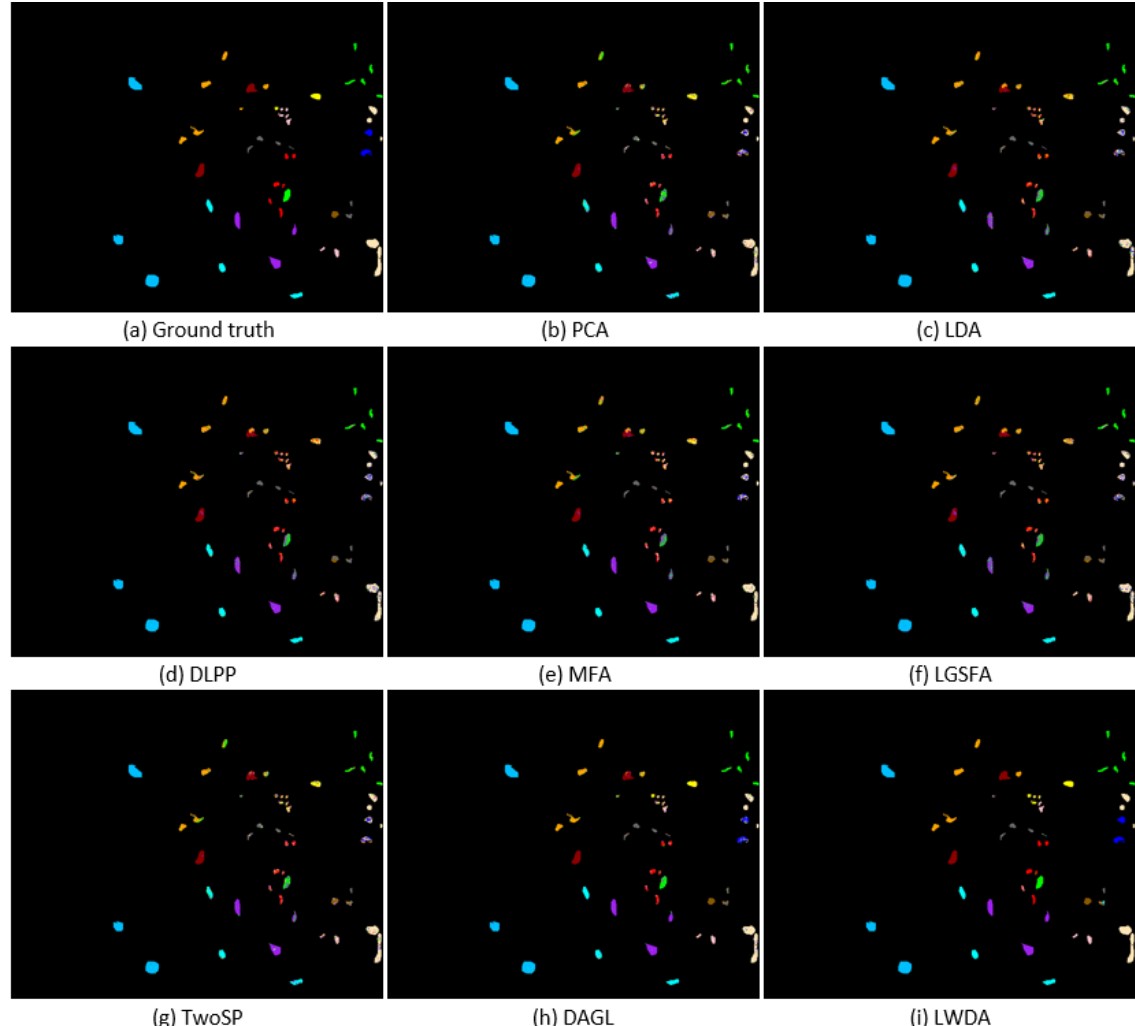

**Figure 4.** Classification maps of different dimensionality reduction methods with NN classifier on the *KSC* dataset ($\tau = 0.05$). (**a**) Ground truth; (**b**) PCA; (**c**) LDA; (**d**) DLPP; (**e**) MFA; (**f**) LGSFA; (**g**) TwoSP; (**h**) DAGL; (**i**) proposed LWDA.

Furthermore, a McNemar test [32,33] was conducted by pairwise comparison to validate the effectiveness of different methods. In the McNemar test, the threshold of significance is set to 0.05. The results of the McNemar test are shown in Table 4, where the methods in the horizontal direction are denoted as the test methods while those in the vertical direction are marked as the reference methods. When the value is smaller than zero, it indicates that the classification performance of the test method is better than that of the reference method; otherwise, the reference method outperforms the test method. Moreover, when the absolute value is larger than 1.96, the two methods have obvious differences. Compared with all the competitors, the absolute values in the LWDA column are larger than 10, which demonstrates that LWDA has a distinct advantage.

**Table 4.** McNemar test of different methods on the *Indian Pines* and *KSC* datasets.

| Methods | Indian Pines | | | | | | | | KSC | | | | | | | |
|---|---|---|---|---|---|---|---|---|---|---|---|---|---|---|---|---|
| | PCA | LDA | DLPP | MFA | LGSFA | TwoSP | DAGL | LWDA | PCA | LDA | DLPP | MFA | LGSFA | TwoSP | DAGL | LWDA |
| RAW | −0.3 | −1.9 | −4.9 | −7.6 | −8.5 | −18.9 | −28.0 | −34.9 | 1.3 | −4.5 | −2.7 | −5.5 | −3.9 | −6.4 | −21.5 | −27.6 |
| PCA | - | −1.7 | −4.7 | −7.3 | −8.2 | −18.4 | −28.6 | −34.8 | - | −4.7 | −2.9 | −5.7 | −4.1 | −6.5 | −21.5 | −27.7 |
| LDA | - | - | −3.6 | −5.6 | −7.5 | −15.9 | −31.1 | −33.4 | - | - | 2.6 | −1.0 | 0.9 | −5.3 | −19.3 | −24.8 |
| DLPP | - | - | - | −2.4 | −4.1 | −13.3 | −26.1 | −31.0 | - | - | - | −3.4 | −1.7 | −6.4 | −20.2 | −25.7 |
| MFA | - | - | - | - | −1.5 | −11.7 | −24.0 | −29.6 | - | - | - | - | 1.8 | −5.1 | −19.8 | −24.2 |
| LGSFA | - | - | - | - | - | −10.0 | −23.7 | −28.7 | - | - | - | - | - | −7.4 | −21.3 | −25.0 |
| TwoSP | - | - | - | - | - | - | −15.5 | −21.4 | - | - | - | - | - | - | −12.5 | −21.0 |
| DAGL | - | - | - | - | - | - | - | −17.7 | - | - | - | - | - | - | - | −10.9 |

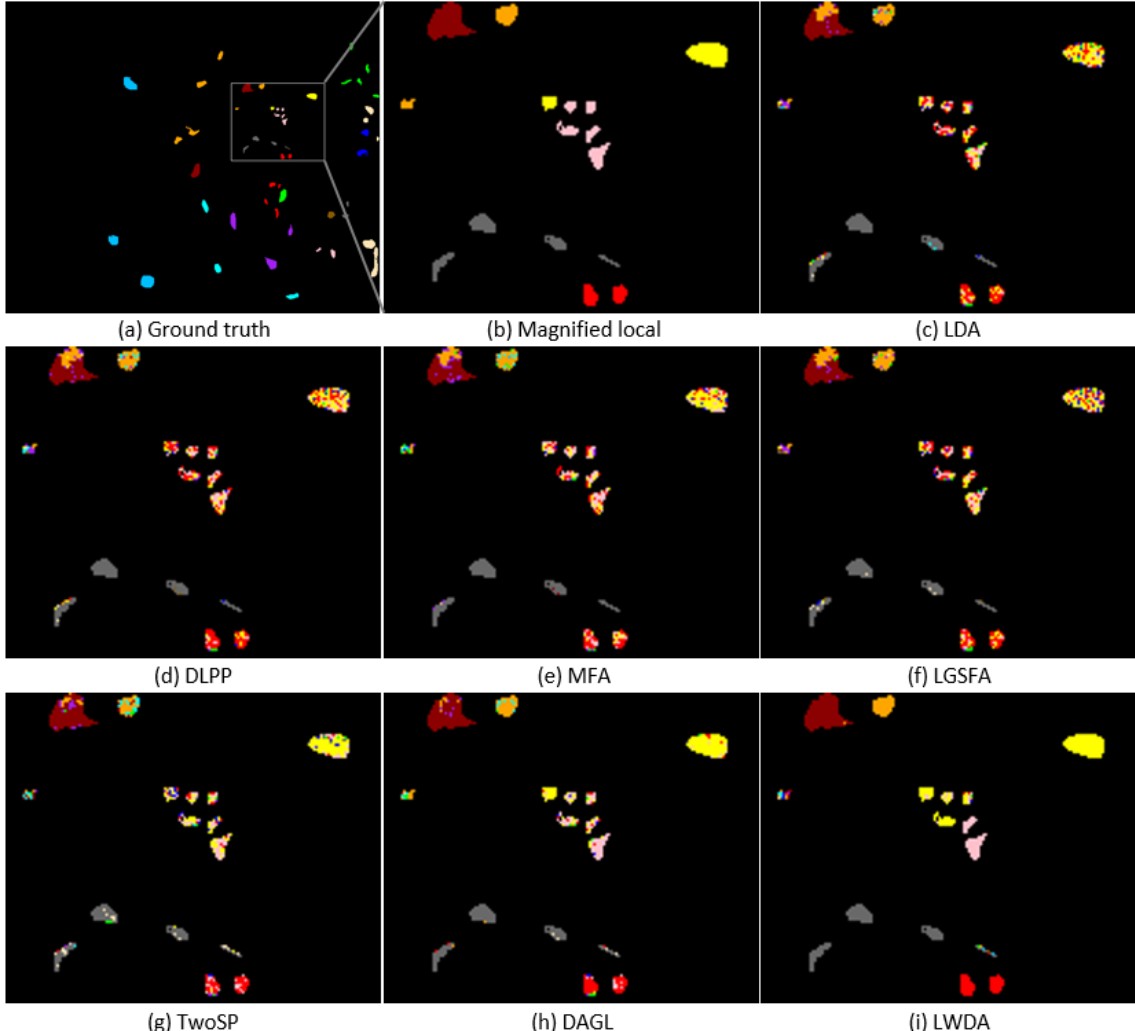

**Figure 5.** Local magnification (with a magnification factor of 4) of the output classification maps shown in Figure 4. (**a**) Selected local region; (**b**) magnified local region; (**c**) LDA; (**d**) DLPP; (**e**) MFA; (**f**) LGSFA; (**g**) TwoSP; (**h**) DAGL; (**i**) proposed LWDA.

### 4.3. Analysis of Computational Cost

**T1** is denoted as the running time of the construction of projection matrix. **T2** and **T3** are defined as the classification times obtained by the NN and SVM classifiers in the testing process.

Table 5 shows **T1**, **T2**, and **T3** in terms of seconds of the different methods. The low dimensionality of the embedding features can reduce the running time in the classification process. To obtain the optimal projection matrix, the proposed LWDA method estimates the class information for each test sample. In addition, LWDA takes the most running time to extract the spectral-spatial information in the discriminant analysis for each training sample. So, the computation of dimensionality reduction in LWDA is larger than the others, excluding TwoSP because it involves a large kernel matrix computation. Compared with **T3**, the running time using the NN classifier is smaller, which also illustrates that the NN classifier for the classification process has a better advantage. LWDA with the SVM classifier needs to construct an SVM model for each test sample, so **T3** of LWDA is much larger than the baselines. Therefore, the NN classifier was applied in the experiments.

**Table 5.** Average computational time (unit: second) of different methods with NN classifier on the *Indian Pines* and *KSC* datasets using $\tau = 0.05$.

| Time | Methods | RAW | PCA | LDA | DLPP | MFA | LGSFA | TwoSP | DAGL | LWDA |
|------|---------|-----|-----|-----|------|-----|-------|-------|------|------|
| *Indian Pines* | **T1** | - | 0.7 | 0.2 | 0.6 | 1.3 | 1.3 | 428.3 | 588.1 | 229.3 |
| | **T2** | 1.7 | 0.7 | 0.6 | 0.7 | 0.6 | 0.7 | 0.7 | 38.1 | 36.9 |
| | **T3** | 112.8 | 22.2 | 20.1 | 21.7 | 23.3 | 25.8 | 27.9 | 7145.3 | 48,347.3 |
| *KSC* | **T1** | - | 0.3 | 0.1 | 0.2 | 0.5 | 0.4 | 89.4 | 218.7 | 157.5 |
| | **T2** | 0.4 | 0.2 | 0.2 | 0.2 | 0.2 | 0.2 | 0.2 | 16.0 | 14.0 |
| | **T3** | 28.4 | 7.0 | 6.2 | 7.5 | 10.1 | 8.4 | 8.3 | 3598.6 | 19,316.0 |

### 4.4. Analysis of Reduced Dimensionality

The optimal reduced dimensionality of each method is discussed in this section. Figure 6 shows the curves of OAs varying with different dimensionalities of the projection matrix, from 2 to 50, for the *Indian Pines* and *KSC* datasets.

Figure 6 demonstrates that the proposed LWDA method achieves the highest OAs constantly. In particular, LWDA exceeds the baselines to a large extent when the dimensionality is less than 5. Furthermore, the classification performance of all the methods becomes stable or decreases when the dimensionality increases to a certain value, which also indicates that a low-dimensional feature subspace is sufficient for preserving the valuable information of HSI data.

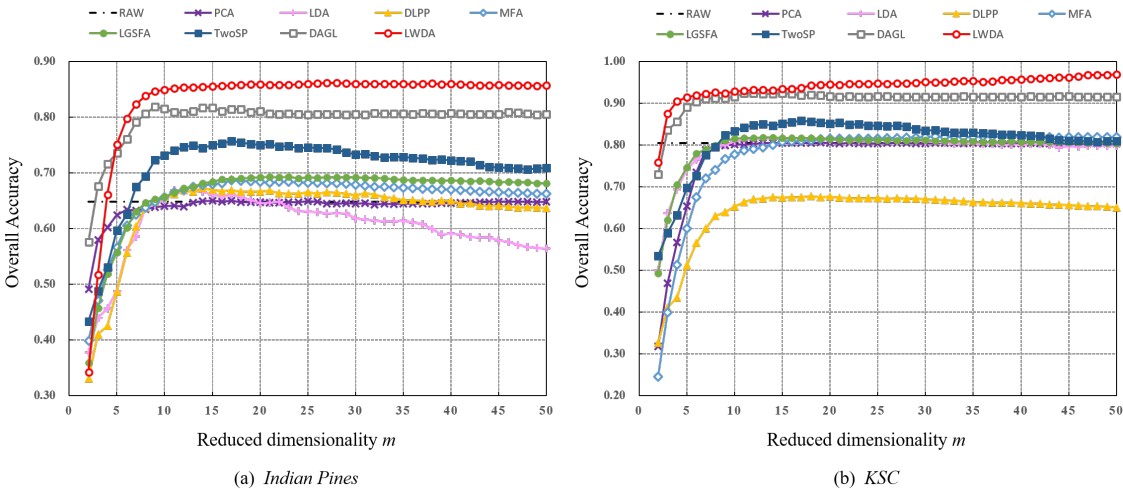

**Figure 6.** Overall classification accuracy (OA) versus the reduced dimensionality of different methods with NN classifier on the (**a**) *Indian Pines* and (**b**) *KSC* datasets using $\tau = 0.05$.

### 4.5. Analysis of Classifier

To evaluate the classification performance of each method with two different classifiers, i.e., NN and SVM, the experiments were repeated five times. For SVM, the LibSVM Toolbox in a MATLAB version was applied with a radial basis function (RBF) kernel [34]. Once the projected features were obtained by each method, the NN and SVM classifiers were applied for the classification process, respectively. Figure 7 shows the classification results obtained by different methods with NN and SVM classifiers on two HSI datasets.

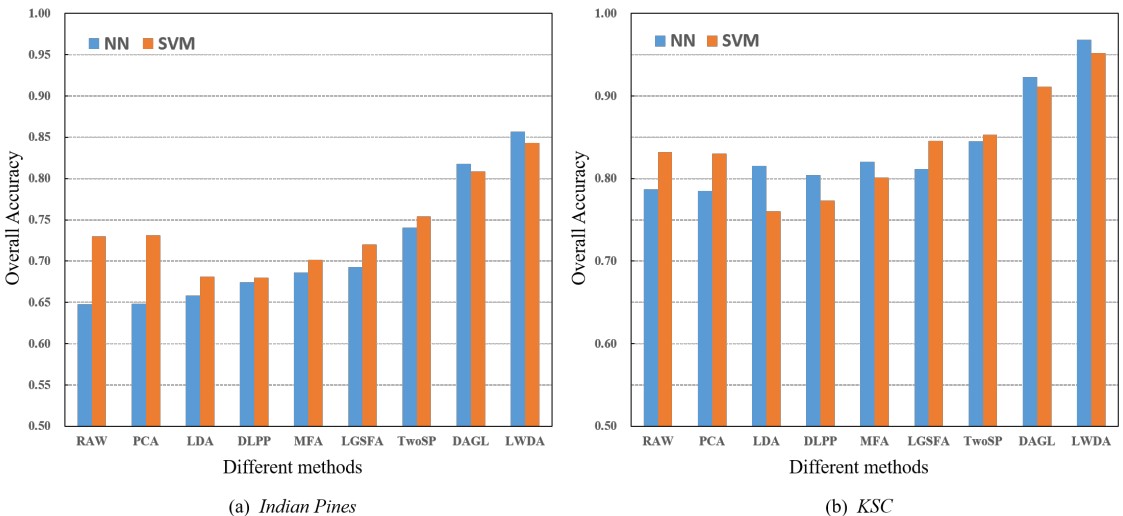

(a) *Indian Pines*        (b) *KSC*

**Figure 7.** OAs obtained by different methods with the NN and support vector machine (SVM) classifiers on the (**a**) *Indian Pines* and (**b**) *KSC* datasets using $\tau = 0.05$.

Figure 7 illustrates that the proposed LWDA method with the NN classifier presents the best classification performance compared with the other dimensionality reduction methods. For most cases on the two dataset, the results with the NN classifier are superior to those with SVM. To unify the classifier in the classification process, the NN classifier was used in all the experiments.

### 4.6. Analysis of Parameters

The proposed LWDA method has two trade-off parameters $r$ and $\beta$. The value of $r$ affects the number of neighbors in the spatial space, while the value of $\beta$ balances the contribution between the weighted scatter matrix model and the spatial consistency matrix. $r$ was tuned with the set $\{3, 5, 7, 9, 11, 13, 15, 17, 19, 21, 23, 25, 27\}$, and $\beta$ was varied with the set $\{0, 0.001, 0.005, 0.01, 0.02, 0.04, 0.05, 0.06, 0.08, 0.1, 0.5, 1, 5, 10, 50, 100\}$. Figure 8 shows the average OAs with respect to the values of parameters $r$ and $\beta$.

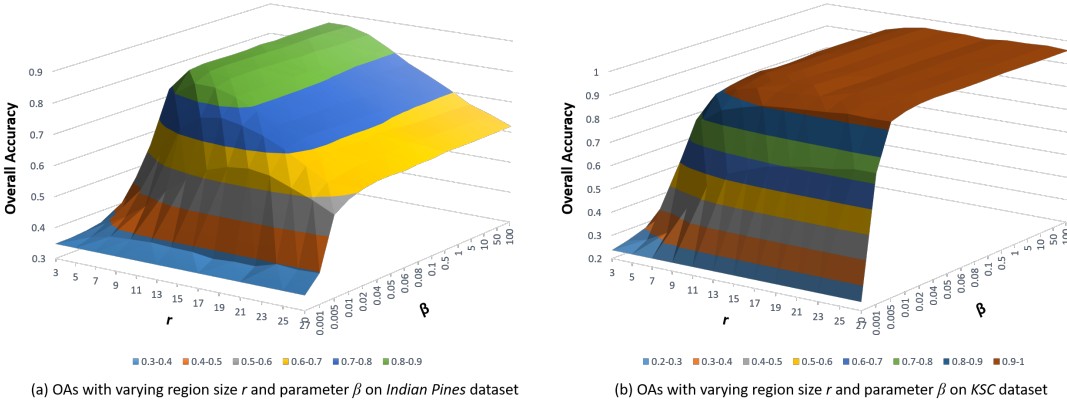

(a) OAs with varying region size *r* and parameter *β* on *Indian Pines* dataset     (b) OAs with varying region size *r* and parameter *β* on *KSC* dataset

**Figure 8.** OAs versus the value of parameters $r$ and $\beta$ in the proposed LWDA method with NN classifier on the (**a**) *Indian Pines* and (**b**) *KSC* datasets using $\tau = 0.05$.

According to Figure 8, when the value of $\beta$ increases, the OAs display a subtle change with a fixed $r$. That is because the spatial consistency matrix generates a similar contribution for the *Indian Pines* and *KSC* datasets. An increased $r$ leads to introducing more between-class samples in the construction of a spatial consistency matrix. For the *Indian Pines* dataset, a peak value is generated in the curved surface map when the value of $r$ reaches 11, and then the OAs begin to slowly decrease when the value

of $r$ continues to increase. If $r$ is a large value, the contribution of the preservation of the local manifold structure will be reduced. Similarly, a peak value of the curved surface is obtained when $r$ increases to 25 for the *KSC* dataset. Therefore, the parameters $r$ and $\beta$ were set to 11 and 0.05 for the *Indian Pines* dataset and 25 and 0.04 for the *KSC* dataset in the experiments.

The proposed method can be divided into two versions: online and offline. The online version of LWDA needs to construct the optimal projection matrix for each test sample, which leads to a large computational cost. To reduce the computational time, the experiments in this paper used the offline version. Figure 9 shows the histograms of the spatial distance between the input test sample and its nearest training sample. The two histograms for the *Indian Pines* and *KSC* datasets illustrate that the spatial distance is mainly distributed in the range of $[1, 5]$. The samples in a small neighborhood should be close to each other in the desired feature subspace. Furthermore, the experimental results shown in the Section 4.2 demonstrate that the offline version of the proposed method achieves better classification performance than the existing approaches.

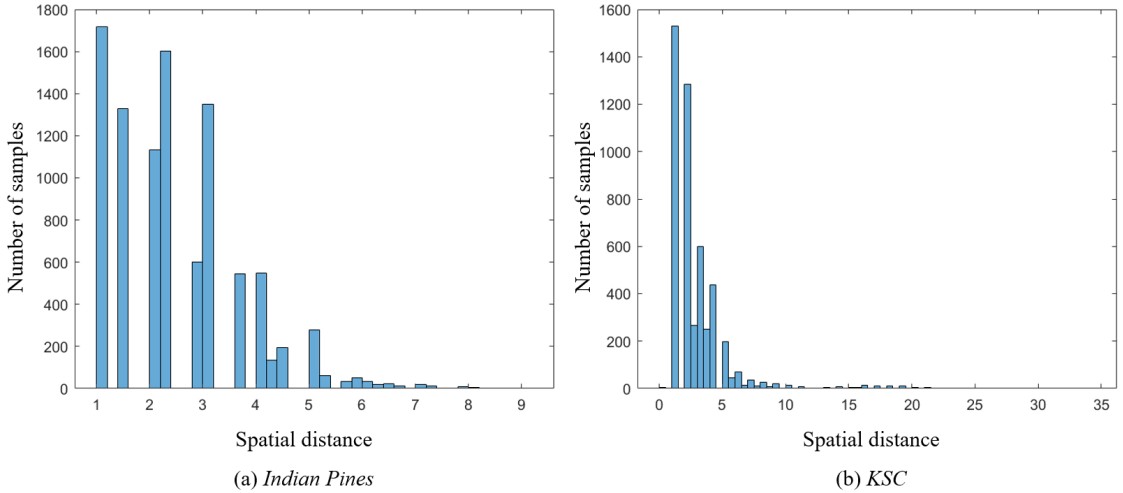

(a) *Indian Pines*        (b) *KSC*

**Figure 9.** Histogram of spatial distance for the (**a**) *Indian Pines* and (**b**) *KSC* datasets.

### 4.7. Analysis of Data Partition Factor $\tau$

The data partition factor $\tau$ affects the number of selected samples in the training process. The influence of different values of $\tau$ was investigated, and results are presented in this section. The value of $\tau$ was tuned with the set $\{0.05, 0.06, 0.07, 0.08, 0.09, 0.1\}$. Tables 6 and 7 show the classification results in terms of average OAs, average KCs, and their STDs with the *Indian Pines* and *KSC* datasets.

In Tables 6 and 7, the OAs and KCs improve with the increase in the data partition factor for all the methods on the two HSI datasets. It implies that a large number of training samples contain more valuable information in the feature representation. DLPP achieves better classification results than PCA and LDA, because DLPP applies the discrimination information to preserve the local structure. TwoSP shows better OAs and KCs than MFA and LGSFA in most conditions, since TwoSP effectively alleviates the nonlinear problem in HSI data and simultaneously preserves the global and local structures. In all experiments, the proposed LWDA method achieves the best classification results under different values of $\tau$, especially with a small $\tau$ value, which indicates that a small number of training samples is enough for a good performance. Moreover, LWDA applies the spectral-spatial information to the discriminant analysis process, which largely enhances the discriminating power of low-dimensional embedding features.

**Table 6.** Classification results (%) with different values of $\tau$ on the *Indian Pines* dataset with NN classifier. Each method has two rows, where the first row is the OA $\pm$ standard deviation (STD) and the second row is the kappa coefficient (KC) $\pm$ STD.

| Methods | $\tau = 0.05$ | $\tau = 0.06$ | $\tau = 0.07$ | $\tau = 0.08$ | $\tau = 0.09$ | $\tau = 0.1$ |
|---|---|---|---|---|---|---|
| RAW | $64.8 \pm 1.0$ | $65.5 \pm 0.7$ | $66.1 \pm 0.4$ | $67.0 \pm 0.5$ | $67.3 \pm 0.5$ | $67.7 \pm 0.4$ |
|  | $59.7 \pm 1.1$ | $60.6 \pm 0.7$ | $61.3 \pm 0.5$ | $62.3 \pm 0.6$ | $62.7 \pm 0.6$ | $63.1 \pm 0.5$ |
| PCA | $64.9 \pm 0.8$ | $65.6 \pm 0.5$ | $66.2 \pm 0.3$ | $67.2 \pm 0.5$ | $67.5 \pm 0.3$ | $67.7 \pm 0.4$ |
|  | $59.8 \pm 0.8$ | $60.7 \pm 0.5$ | $61.5 \pm 0.3$ | $62.5 \pm 0.6$ | $62.9 \pm 0.4$ | $63.1 \pm 0.5$ |
| LDA | $65.8 \pm 1.7$ | $67.9 \pm 0.8$ | $69.7 \pm 1.2$ | $71.0 \pm 0.6$ | $71.8 \pm 0.9$ | $72.6 \pm 1.2$ |
|  | $60.8 \pm 2.1$ | $63.1 \pm 0.9$ | $65.3 \pm 1.4$ | $66.9 \pm 0.7$ | $67.7 \pm 1.0$ | $68.7 \pm 1.3$ |
| DLPP | $67.5 \pm 0.9$ | $70.3 \pm 0.5$ | $72.8 \pm 0.8$ | $73.9 \pm 0.5$ | $75.5 \pm 0.6$ | $76.7 \pm 1.1$ |
|  | $62.9 \pm 1.0$ | $66.0 \pm 0.6$ | $68.9 \pm 0.9$ | $70.2 \pm 0.6$ | $72.0 \pm 0.7$ | $73.3 \pm 1.3$ |
| MFA | $68.6 \pm 1.4$ | $71.6 \pm 1.6$ | $73.5 \pm 1.8$ | $74.8 \pm 1.5$ | $76.6 \pm 1.1$ | $77.7 \pm 1.2$ |
|  | $63.9 \pm 1.5$ | $67.3 \pm 1.8$ | $69.6 \pm 2.1$ | $71.1 \pm 1.7$ | $73.2 \pm 1.3$ | $74.4 \pm 1.4$ |
| LGSFA | $69.3 \pm 1.1$ | $71.6 \pm 1.2$ | $73.4 \pm 1.0$ | $75.0 \pm 1.0$ | $76.6 \pm 0.6$ | $77.1 \pm 1.0$ |
|  | $64.7 \pm 1.3$ | $67.4 \pm 1.4$ | $69.5 \pm 1.2$ | $71.3 \pm 1.2$ | $73.2 \pm 0.8$ | $73.8 \pm 1.2$ |
| TwoSP | $73.9 \pm 1.2$ | $75.3 \pm 0.8$ | $76.3 \pm 1.2$ | $76.9 \pm 0.6$ | $77.7 \pm 1.1$ | $78.6 \pm 0.9$ |
|  | $70.1 \pm 1.4$ | $71.8 \pm 0.9$ | $72.9 \pm 1.4$ | $73.6 \pm 0.7$ | $74.5 \pm 1.2$ | $75.6 \pm 1.1$ |
| DAGL | $81.7 \pm 1.2$ | $83.5 \pm 1.3$ | $84.1 \pm 1.0$ | $85.2 \pm 0.9$ | $85.7 \pm 1.0$ | $86.5 \pm 1.2$ |
|  | $79.2 \pm 1.5$ | $81.1 \pm 1.5$ | $82.8 \pm 1.3$ | $83.9 \pm 1.3$ | $84.5 \pm 0.9$ | $85.1 \pm 0.8$ |
| LWDA | $85.1 \pm 1.0$ | $87.0 \pm 1.6$ | $87.7 \pm 1.1$ | $88.5 \pm 1.0$ | $88.9 \pm 1.2$ | $89.3 \pm 0.7$ |
|  | $83.0 \pm 1.1$ | $85.2 \pm 1.9$ | $86.0 \pm 1.2$ | $86.9 \pm 1.2$ | $87.3 \pm 1.3$ | $87.8 \pm 0.7$ |

**Table 7.** Classification results (%) with different values of $\tau$ on the *KSC* dataset with NN classifier. Each method has two rows, where the first row is the OA $\pm$ STD and the second row is the KC $\pm$ STD.

| Methods | $\tau = 0.05$ | $\tau = 0.06$ | $\tau = 0.07$ | $\tau = 0.08$ | $\tau = 0.09$ | $\tau = 0.1$ |
|---|---|---|---|---|---|---|
| RAW | $79.7 \pm 0.6$ | $79.8 \pm 0.9$ | $80.3 \pm 0.4$ | $81.1 \pm 0.4$ | $82.0 \pm 0.4$ | $82.2 \pm 0.4$ |
|  | $77.2 \pm 0.6$ | $77.2 \pm 1.0$ | $78.1 \pm 0.4$ | $78.9 \pm 0.4$ | $79.9 \pm 0.5$ | $80.2 \pm 0.5$ |
| PCA | $79.5 \pm 0.6$ | $79.4 \pm 0.7$ | $80.2 \pm 0.3$ | $80.9 \pm 0.4$ | $81.9 \pm 0.4$ | $82.0 \pm 0.5$ |
|  | $77.2 \pm 0.6$ | $77.0 \pm 0.8$ | $78.0 \pm 0.3$ | $78.8 \pm 0.4$ | $79.8 \pm 0.5$ | $80.0 \pm 0.6$ |
| LDA | $81.4 \pm 0.6$ | $84.3 \pm 1.2$ | $85.8 \pm 1.1$ | $86.7 \pm 1.0$ | $87.8 \pm 0.7$ | $88.3 \pm 1.0$ |
|  | $79.3 \pm 0.7$ | $82.5 \pm 1.3$ | $84.1 \pm 1.2$ | $85.2 \pm 1.1$ | $86.4 \pm 0.8$ | $86.9 \pm 1.1$ |
| DLPP | $80.3 \pm 0.6$ | $84.1 \pm 1.0$ | $85.6 \pm 0.7$ | $86.8 \pm 1.1$ | $87.7 \pm 0.9$ | $88.0 \pm 1.0$ |
|  | $78.1 \pm 0.7$ | $82.3 \pm 1.1$ | $83.9 \pm 0.7$ | $85.3 \pm 1.2$ | $86.3 \pm 1.0$ | $86.6 \pm 1.1$ |
| MFA | $82.0 \pm 0.9$ | $84.1 \pm 0.5$ | $85.3 \pm 0.5$ | $85.9 \pm 0.6$ | $86.3 \pm 0.8$ | $86.9 \pm 0.5$ |
|  | $80.0 \pm 1.0$ | $82.3 \pm 0.5$ | $83.6 \pm 0.6$ | $84.3 \pm 0.7$ | $84.7 \pm 0.9$ | $85.4 \pm 0.6$ |
| LGSFA | $81.2 \pm 1.5$ | $84.4 \pm 1.1$ | $86.5 \pm 0.9$ | $87.6 \pm 1.2$ | $88.7 \pm 1.1$ | $89.5 \pm 0.7$ |
|  | $79.0 \pm 1.7$ | $82.7 \pm 1.2$ | $85.0 \pm 0.9$ | $86.2 \pm 1.3$ | $87.4 \pm 1.3$ | $88.3 \pm 0.8$ |
| TwoSP | $84.5 \pm 0.6$ | $85.3 \pm 1.2$ | $86.8 \pm 1.0$ | $88.0 \pm 1.0$ | $89.0 \pm 1.0$ | $90.1 \pm 1.0$ |
|  | $83.3 \pm 0.8$ | $84.2 \pm 1.1$ | $85.4 \pm 1.0$ | $86.4 \pm 1.0$ | $87.8 \pm 1.1$ | $88.9 \pm 1.1$ |
| DAGL | $92.3 \pm 0.7$ | $92.7 \pm 1.3$ | $93.3 \pm 1.1$ | $93.7 \pm 0.9$ | $93.9 \pm 0.6$ | $94.0 \pm 0.5$ |
|  | $91.4 \pm 0.9$ | $92.0 \pm 1.2$ | $92.9 \pm 1.3$ | $93.4 \pm 1.0$ | $93.6 \pm 0.7$ | $93.8 \pm 0.6$ |
| LWDA | $96.8 \pm 0.7$ | $97.2 \pm 1.1$ | $97.9 \pm 1.3$ | $98.3 \pm 0.8$ | $98.4 \pm 0.7$ | $98.5 \pm 0.3$ |
|  | $96.4 \pm 0.8$ | $96.9 \pm 1.2$ | $97.7 \pm 1.4$ | $98.1 \pm 0.9$ | $98.2 \pm 0.8$ | $98.4 \pm 0.4$ |

## 5. Conclusions

In this paper, a new supervised dimensionality reduction method, named LWDA, is proposed on the basis of the spectral-spatial information of HSI data. During the discriminant analysis, LWDA uses the proposed weighted scatter matrix model and computes the spatial consistency matrix for each data sample, which can adaptively learn local manifold structures of the original HSI data. In addition, LWDA preserves the within-class properties and suppresses the between-class characteristics in an optimal low-dimensional feature subspace.

Through the experiments on two real-world HSI datasets, i.e., *Indian Pines* and *KSC*, LWDA achieves better classification performance than the existing dimensionality reduction approaches.

In particular, a small data portion of the training set is enough for a satisfactory classification performance. The overall accuracy obtained by LWDA increases by at least 17% in comparison with RAW when the data partition factor is 0.05. In addition, the McNemar test demonstrates that LWDA has statistical significance when compared with the baselines. For LWDA, the absolute value of the McNemar test is at least $10 > 1.96$. LWDA learns similarity relationships of the within-class samples and the means of different classes, as well as creates more available information in the subsequent classification. Hence, LWDA achieves the qualitative and quantitative results in the experiments.

Our future work will focus on how to extend the online version of the proposed method to quickly represent the spectral-spatial information and improve the computational efficiency.

**Author Contributions:** All the authors designed the study and discussed the basic structure of the manuscript. X.L. carried out the experiments and finished the first version of this manuscript. L.Z. provided the framework of the experiments. J.Y. reviewed and edited this paper.

**Funding:** This work was supported in part by the National Natural Science Foundation of China under Grants 61601416, 61771349, 61711530239 and 41701417, the Fundamental Research Funds for the Central Universities, China University of Geosciences, Wuhan, under Grant CUG170612, the Hong Kong Scholars Program under Grant XJ2017030, and the Hong Kong Polytheistic University Research Fund.

**Conflicts of Interest:** All the authors declare no conflict of interest.

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
