# Peer review of "Locally Weighted Discriminant Analysis for Hyperspectral Image Classification"

_remotesensing, doi:10.3390/rs11020109_

Round 1

Reviewer 1 Report

This paper proposes a novel weighted linear discriminant analysis based dimensionality reduction method for hyperspectral data. The proposed model takes spatial information into account which can be important for Hyperspectral Data Classification when context provides useful information. Below, you can find my comments about the paper.

The authors should fix some typos and grammar mistakes. For example, in line 127 it says `which will limits`. There are some other grammar mistakes in Figure 1 and rest of the paper.

The authors should describe what x and y axes are in Figure 1. Also, the figure 1 should be briefly described in the caption section.

In Algorithm 1, at the end of the for loop, the authors should add a line as P_i* = P_i.

In Table 4, the authors should highlight the unit of the computational time. Is it second or millisecond?

One major concern about the paper is that it relies on nearest training sample's projection matrix to perform dimensionality reduction. What if the spatially closest training sample is far away to the test sample of interest? I have a feeling that the weighted LDA is specifically designed to improve the HSI classification on the Indian Pines dataset. For example, in this direction, the authors can experiment with the increasing spatial distance between test and the nearest training sample to measure the effect of how important the spatial distance is.

In equation 14, the authors use the notation z, however it is not described in the text. What is the difference between zi and zj? Is it element-wise difference between the original vectors?

The paper only considers relatively old methods when performing comparison between their method and others. More recent approaches should be included in the experiments section.

The authors should justify why SVM performs very poorly on the Indian Pines dataset while it performs similarly to Nearest Neighbor method in KSC dataset.

Reviewer 2 Report

Review report on manuscript number ID Remote Sensing-405184 submitted to "Remote Sensing".

Title: Locally Weighted Discriminant Analysis for Hyperspectral Image Classification

Authors: Xiaoyan Li, Lefei Zhang and Jane You

This paper proposes a new supervised dimensionality reduction method, named as LWDA, on the basis of the spectral-spatial information of HSI data. In the discriminant analysis, LWDA presents a weighted scatter matrix model and a spatial consistency matrix to adaptively learn local manifold structures of the original HSI data. The subject is interesting and relevant to the field of this journal.

A number of issues in this paper can be listed as follows:

Generally: The language should be further improved by a native English speaker.

Abstract: The abstract does not provide the reader with information about the results obtained. It has not any numerical values. It needs to be improved, giving more numerical values for the results.

The authors should change all the sentences in the text which are written in the first plural (line 60, 76, 80, 103, 133, 149, 175, 191, 218, 221, 223, 230, 244, 251, 263, 265, 268, 275, 276, 279, 282, 296, 297 and 300.  

In lines 206 to 207, the authors mention that “Setting t to 0.05, the number of samples that are chosen for training is [1428*0.05] = 72, while the remaining 1356 samples are used for testing. Why the authors decide this samples ratio?  According the literature the training sample is bigger than testing sample (80/20 or 70/30 ratio). Did the authors tried different data ratio? I am quite sure that the classification results will be differently if the authors change the ratio.

Conclusion: The Conclusion is not suitable, should give more useful conclusions. This section presented in brief. This paragraph should be reformed. Should include numerical values for the results.

Reviewer 3 Report

authors have presented a nice algorithm to illustrate classification of HSI data, while they have some shortcoming in the manuscript. 

Major- 

How did you validated the accuracy results? I suggest you to implement bootstraping method for validation of accuracy assessment (as the performance of the accuracy assessment is influenced by several aspects). 

See Foody et al on the accuracy assessment topics for precise results. 

1. English improvement is required at several places. 

2. Figure 4, 5 and 7 needs improvement. 

3. Reason  should be provided - why SVM is giving less accuracy in case in indian pine classification, as compared to other while NN is giving very much higher accuracy (a jump from 0.74 from second highest to 0.86 for indian pine ?

4. validation sample has not been illustrated in the manuscript.

Round 2

Reviewer 2 Report

The authors responded to all my questions and took into account all my suggestions.

Reviewer 3 Report

Manuscript can be accepted for publication.